# Structure and evolution of alanine/serine decarboxylases and the engineering of theanine production

**Hao Wang[1†], Biying Zhu[2†], Siming Qiao[2], Chunxia Dong[2], Xiaochun Wan[2], Weimin Gong[1]\*, Zhaoliang Zhang[2]\***

[1]Department of Life Sciences and Medicine, University of Science and Technology of China, Hefei, China; [2]State Key Laboratory of Tea Plant Biology and Utilization, Anhui Agricultural University, Hefei, China

**Abstract** Ethylamine (EA), the precursor of theanine biosynthesis, is synthesized from alanine decarboxylation by alanine decarboxylase (AlaDC) in tea plants. AlaDC evolves from serine decarboxylase (SerDC) through neofunctionalization and has lower catalytic activity. However, lacking structure information hinders the understanding of the evolution of substrate specificity and catalytic activity. In this study, we solved the X-ray crystal structures of AlaDC from *Camellia sinensis* (CsAlaDC) and SerDC from *Arabidopsis thaliana* (AtSerDC). Tyr[341] of AtSerDC or the corresponding Tyr[336] of CsAlaDC is essential for their enzymatic activity. Tyr[111] of AtSerDC and the corresponding Phe[106] of CsAlaDC determine their substrate specificity. Both CsAlaDC and AtSerDC have a distinctive zinc finger and have not been identified in any other Group II PLP-dependent amino acid decarboxylases. Based on the structural comparisons, we conducted a mutation screen of CsAlaDC. The results indicated that the mutation of L110F or P114A in the CsAlaDC dimerization interface significantly improved the catalytic activity by 110% and 59%, respectively. Combining a double mutant of CsAlaDC[L110F/P114A] with theanine synthetase increased theanine production 672% in an *in vitro* system. This study provides the structural basis for the substrate selectivity and catalytic activity of CsAlaDC and AtSerDC and provides a route to more efficient biosynthesis of theanine.

## eLife assessment

This study reports a comparative biochemical and structural analysis of two PLP decarboxylase enzymes from plants. The work is **useful** because of the potential application of these enzymes in industrial theanine production. The structure provides a **solid** basis for understanding substrate specificity but some aspects of the work are **incomplete**. The paper will be of interest to enzymologists studying PLP enzymes and those working on enzyme engineering in plants.

## Introduction

Tea is one of the most popular natural non-alcoholic beverages consumed worldwide. About 2 billion cups of tea are consumed daily (*Drew, 2019*). Theanine (γ-glutamylethylamide) is a unique non-protein amino acid and the most abundant free amino acid in tea plants (*Camellia sinensis*). It accounts for more than 50% of the total free amino acids and approximately 1–2% of the dry weight of the new shoots of tea plants (*Juneja et al., 1999*). Theanine is the secondary metabolite conferring the umami taste of tea infusion and also balances the astringency and bitterness of tea infusion caused by catechins and caffeine (*Lin et al., 2022*). It has also many health-promoting functions, including neuroprotective effects, enhancement of immune functions, and potential anti-obesity capabilities,

**\*For correspondence:**
wgong@ustc.edu.cn (WG);
zhlzhang@ahau.edu.cn (ZZ)

[†]These authors contributed equally to this work

**Competing interest:** The authors declare that no competing interests exist.

among others (*Juneja et al., 1999*; *Haskell et al., 2008*; *Kakuda et al., 2000*; *Liu et al., 2009*; *Lu et al., 2004*; *Takagi et al., 2010*; *Zheng et al., 2004*). Therefore, theanine content is highly correlated with green tea quality (*Lin et al., 2023*).

Theanine is synthesized from EA and glutamate (Glu) by theanine synthetase (*Deng et al., 2008*; *Deng et al., 2009*). Importantly, the large amount of theanine biosynthesis is determined by the high availability of EA in tea plants (*Cheng et al., 2019*). Therefore, the evolution of EA biosynthesis in tea plants provided the basis for theanine biosynthesis and the formation of tea quality. EA is synthesized from alanine decarboxylation by alanine decarboxylase (CsAlaDC *Takeo, 1974*; *Figure 1A*). Indeed, CsAlaDC expression level and catalytic activity largely determine the theanine accumulation level in tea plants (*Zhu et al., 2021*). As a novel gene, *CsAlaDC* had not been reported in other plant species before it was identified in tea plants. Previous studies indicated that *AlaDC* originated from the *SerDC* by gene duplication in tea plants (*Bai et al., 2021*). However, despite sharing highly conserved amino acid sequences, CsAlaDC and CsSerDC specifically catalyze the decarboxylation of alanine and serine, respectively (*Bai et al., 2021*; *Figure 1A and B*). In addition, CsAlaDC exhibited a much lower enzymatic activity compared with CsSerDC (*Bai et al., 2021*). However, the structural basis and key amino acids underlying the evolution of the substrate specificity and enzymatic activity of CsAlaDC are unknown.

SerDC belongs to the Group II pyridoxal-5-phosphate (PLP)-dependent decarboxylase superfamily (*Han and Shin, 2022*). These Group II amino acid decarboxylases produce many bioactive secondary metabolites and signaling molecules (*Berger et al., 2009*; *Khan and Nawaz, 2016*; *Ng et al., 2015*). In plants, SerDC catalyzes the biosynthesis of ethanolamine from serine, which was first functionally characterized in *Arabidopsis thaliana* (*Rontein et al., 2001 Figure 1A*). Ethanolamine is an important metabolite in the synthesis of phosphatidylethanolamine and phosphatidylcholine, two main phospholipids involved in maintaining eukaryotic membrane structures in eukaryotic cell membranes (*Gibellini and Smith, 2010*; *Mudd and Datko, 1989*; *Zinser et al., 1991*). Furthermore, the study has shown that SerDC is essential in the embryogenesis of *Arabidopsis* (*Yunus et al., 2016*). SerDC is of vital importance for plant growth and development, but the mechanism of substrate recognition and catalytic activity of SerDC also remains unclear.

In this work, we attempt to understand the mechanism of functional divergence of plant AlaDC and SerDC by analyzing their structural characteristics. Here, we obtained the X-ray crystal structure of CsAlaDC and AtSerDC. According to the crystal structure, we found a distinctive zinc finger structure located at both CsAlaDC and AtSerDC, which has not been identified in any other Group II PLP-dependent amino acid decarboxylases that have been previously characterized. By comparing the substrate binding pockets, we identified Phe[106] of CsAlaDC and Tyr[111] of AtSerDC as the crucial sites for substrate specificity. By conducting mutation screening based on the protein structures, we identified the amino acids repressing the catalytic activity and discovered that CsAlaDC[L110F/P114A] exhibited a 2.3-fold increase in catalytic activity compared to that of CsAlaDC and enhanced the engineering of theanine production *in vitro*.

## Results

### Enzymatic properties of CsAlaDC, AtSerDC, and CsSerDC

CsAlaDC originates from CsSerDC by gene duplication and neofunctionalization in tea plants, but they catalyze different metabolic processes (*Figure 1A*). We performed a multiple sequence alignment of the amino acid sequences of the six SerDCs from kiwifruit (*Actinidia chinensis*), grape (*Vitis vinifera*), coffee (*Coffea eugenioides*), cocoa (*Theobroma cacao*), *Arabidopsis* and tea plants, and CsAlaDC (*Figure 1B*). The results showed that the amino acid sequences of CsAlaDC and SerDCs were highly conserved, but CsAlaDC has amino acid mutations in highly conserved regions compared with SerDCs.

Next, to verify the substrate specificity and enzyme activity, we conducted enzyme activity detection and enzyme kinetics analysis for CsAlaDC, CsSerDC, and AtSerDC. The 5′ truncated *CsAlaDC* and *AtSerDC* were inserted into the pET22b and pET28a expression vectors to generate recombinant plasmids pET22b-*CsAlaDC* and pET28a-*AtSerDC*, correspondingly. Additionally, the full-length protein of CsSerDC was inserted into the pET28a expression vector to generate the recombinant plasmid pET28a-*CsSerDC*. Subsequently, the recombinant proteins were expressed in *Escherichia coli*

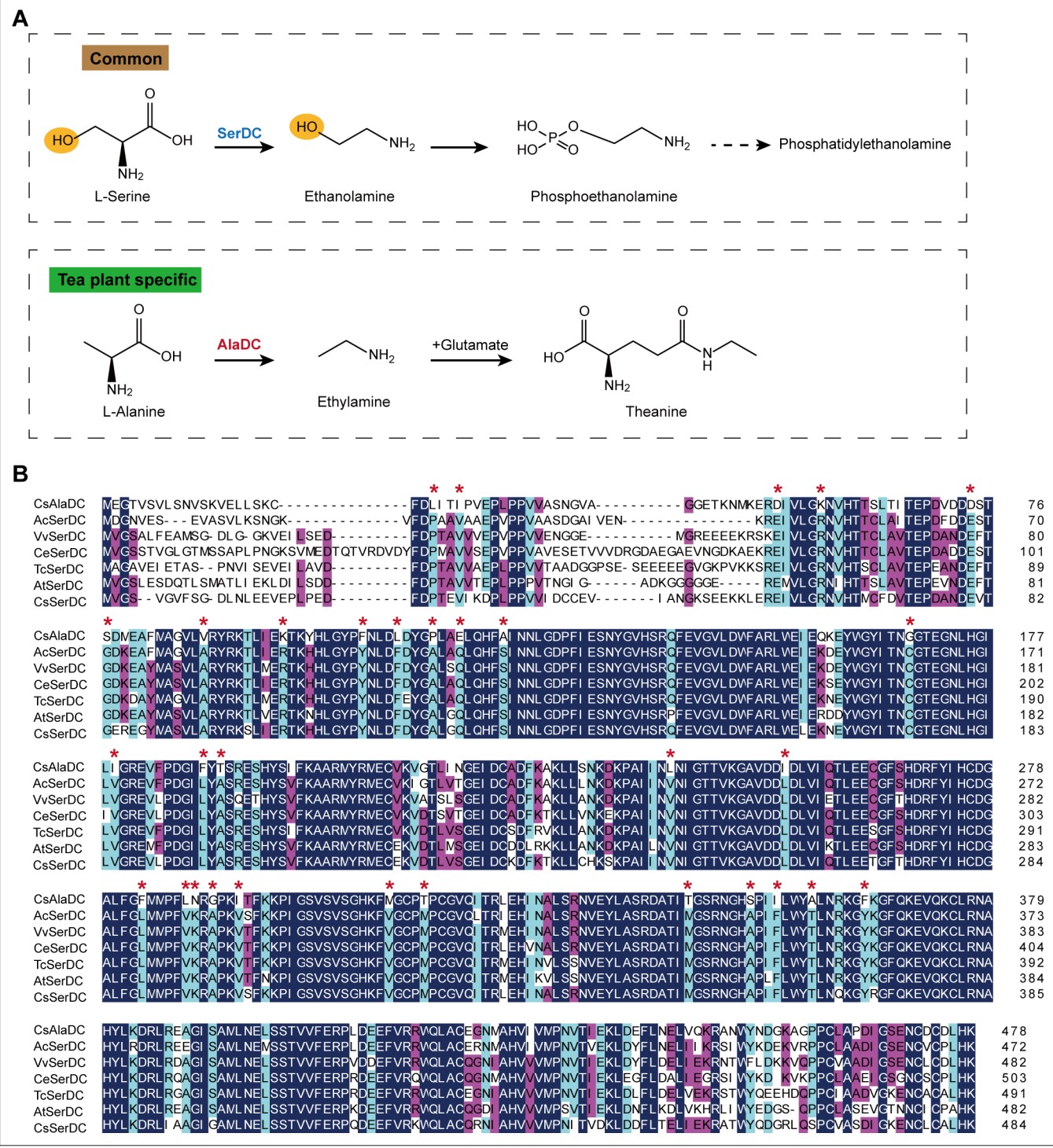

**Figure 1.** Metabolic pathways and sequence analysis of *Camellia sinensis* alanine decarboxylase (CsAlaDC) and SerDCs in plants. (**A**) The decarboxylation of serine and alanine in plants. SerDC, serine decarboxylase; AlaDC, alanine decarboxylase. (**B**) Multiple alignments of the amino acid sequences of AlaDC and SerDCs. The amino acid sequences of the six SerDCs from kiwifruit (*Actinidia chinensis*), grape (*Vitis vinifera*), coffee (*Coffea eugenioides*), cocoa (*Theobroma cacao*), and *Arabidopsis*. Primary (100%), secondary (80%), and tertiary (60%) conserved percent of similar amino acid residues were shaded in deep blue, light blue, and cheer red, respectively. '*' indicated amino acid residues mutated only in CsAlaDC.

and purified via nickel affinity chromatography. The purified proteins were examined through SDS-PAGE analysis (*Figure 2A*), supplemented by size-exclusion chromatography assessments (*Figure 2—figure supplement 1*), it seems that the protein manifests in an oligomeric configuration within the solution.

To verify the substrate specificity of CsAlaDC, AtSerDC, and CsSerDC, enzyme activity assays were conducted using Ala and Ser as substrates, followed by product identification via UPLC analysis. The results showed that CsAlaDC can effectively catalyze the decarboxylation of alanine to generate EA, whereas its catalytic efficacy towards serine was relatively inferior. Conversely, AtSerDC and CsSerDC selectively catalyzed serine decarboxylation to yield ethanolamine but did not exhibit activity towards alanine (*Figure 2B*). These findings confirmed that CsAlaDC and SerDCs do not have promiscuous decarboxylase activity on Ala and Ser.

The kinetic properties of CsAlaDC, AtSerDC, and CsSerDC were determined through the use of corresponding substrates (*Figure 2C*), and the results are presented in *Table 1*. The kinetic parameter $Km$ of CsAlaDC was determined to be 1.215 mM, which is similar to that of AtSerDC and CsSerDC at 1.912 mM and 2.364 mM, respectively. However, AtSerDC has a $Vmax$ of 7.787 µmo L$^{-1}$ s$^{-1}$, which is 4.6-fold that of CsAlaDC's $Vmax$ of 1.709 µmo L$^{-1}$ s$^{-1}$; while CsSerDC has a $Vmax$ of 9.031 µmo L$^{-1}$ s$^{-1}$, which is 5.3-fold that of CsAlaDC. The findings suggest that the enzymatic catalytic efficiency of CsSerDC and AtSerDC is approximately triple compared to that of CsAlaDC, and the $Vmax$ of CsAlaDC is considerably lower than that of both CsSerDC and AtSerDC. The observed disparity in $Vmax$ suggests that CsAlaDC and SerDCs may exhibit discrepancies in substrate-to-product conversion rates under saturated substrate conditions. It is plausible that SerDCs possess a higher turnover rate, facilitating the rapid conversion of substrate to product and subsequent release of the enzyme. Another possibility is that SerDCs demonstrate enhanced stability, thereby maintaining superior catalytic activity even at higher substrate concentrations, consequently leading to a higher $Vmax$ value. Due to the similarity in catalytic activity between CsSerDC and AtSerDC, we chose the more representative AtSerDC for further analysis.

## The overall structures of CsAlaDC and AtSerDC

To enhance our comprehension of CsAlaDC and AtSerDC, we conducted structural analyses of these two proteins. Through optimization of crystallization conditions, we successfully determined the crystal structures of CsAlaDC, CsAlaDC-EA, and AtSerDC at resolutions of 2.50 Å, 2.60 Å, and 2.85 Å, respectively (*Supplementary file 1a*).

The overall structure of CsAlaDC or AtSerDC is homodimeric with two subunits exhibiting an asymmetrical arrangement (*Figure 3A and D*). The monomer of CsAlaDC or AtSerDC is divided into three distinct structural domains: an N-terminal domain (N-terminal –104 aa in CsAlaDC, N-terminal –109 aa in AtSerDC), a large domain (105–364 aa in CsAlaDC, 110–369 aa in AtSerDC), and a small C-terminal domain (365 aa-C-terminus in CsAlaDC, 370-C-terminus as in AtSerDC), colored light pink, khaki, and sky blue, respectively. Compared to the full-length protein, the N-terminal structures of CsAlaDC and AtSerDC were truncated by 60 and 65 amino acid residues, respectively. The N-terminal domain of the truncated protein contains a long α-helix, which is connected to the large domain through a long loop. The helix of one subunit is antiparallel to the corresponding helix in the neighboring subunit, forming a clamp. Therefore, the N-terminal domain may not be an independent folding unit and is likely only stable in dimers. The large domain contains a seven-stranded mixed β-sheet surrounded by eight α-helices, while the C-terminal domain is composed of a three-stranded antiparallel β-sheet and three α-helices (*Figure 3—figure supplement 1*). The catalytic site of the enzyme is positioned within a superficial crevice at the junction of two subunits forming a dimeric arrangement. The amino acid residues derived from both subunits participate in the binding of the PLP, effectively anchoring it to the active site. In the catalytic site, one monomer accommodates PLP, while the other monomer also contributes to PLP binding and enzymatic function (*Figure 3A and D*).

Notably, our investigation has revealed the presence of a distinctive zinc finger structure (as depicted in *Figure 3A and D*) located at the C-terminus of CsAlaDC and AtSerDC. This structure is composed of a loop structure spanning 17 amino acid residues, wherein coordination of Zn$^{2+}$ is facilitated by three Cys residues and one His residue. Importantly, this particular configuration is exclusive to the two proteins under examination and has not been identified in any other Group II

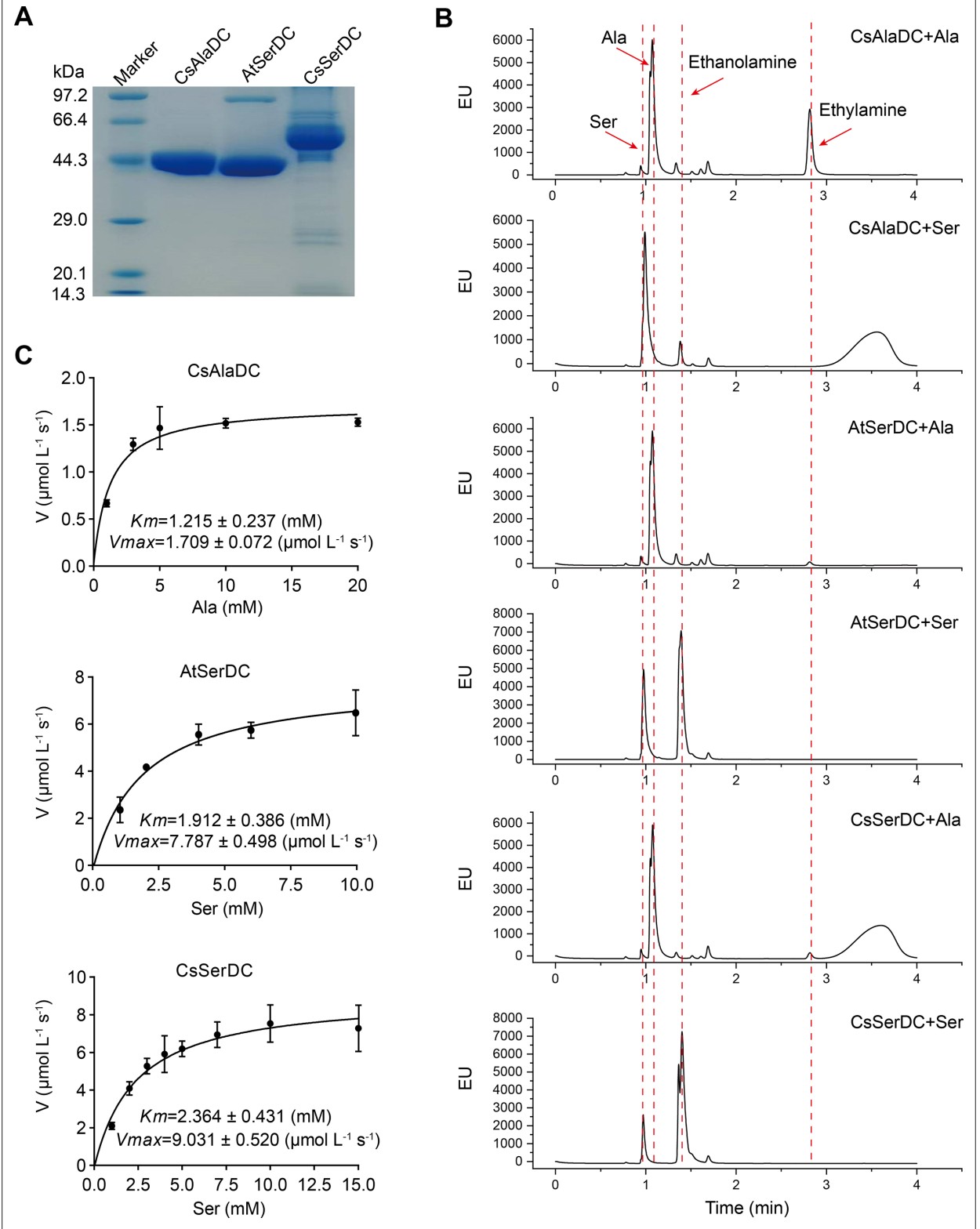

**Figure 2.** Purification and characterization of *Camellia sinensis* alanine decarboxylase (CsAlaDC), *Arabidopsis thaliana* serine decarboxylase (AtSerDC), and *Camellia sinensis* serine decarboxylase (CsSerDC). (**A**) Identification of CsAlaDC, AtSerDC, and CsSerDC using SDS-PAGE. (**B**) Detection of enzyme activities of CsAlaDC, AtSerDC, and CsSerDC by UPLC. (**C**) Reaction rates of substrates with different concentrations catalyzed by CsAlaDC, AtSerDC, and CsSerDC.

The online version of this article includes the following source data and figure supplement(s) for figure 2:

*Figure 2 continued on next page*

*Figure 2 continued*

**Source data 1.** Uncropped and labelled gels for *Figure 2*.

**Source data 2.** Raw unedited gels for *Figure 2*.

**Source data 3.** Raw excel data for *Figure 2C*.

**Figure supplement 1.** Comparison of elution profiles of *Camellia sinensis* alanine decarboxylase (CsAlaDC), *Arabidopsis thaliana* serine decarboxylase (AtSerDC), and *Camellia sinensis* serine decarboxylase (CsSerDC).

**Figure supplement 2.** Purification of truncated *Camellia sinensis* alanine decarboxylase (CsAlaDC) and *Arabidopsis thaliana* serine decarboxylase (AtSerDC).

**Figure supplement 2—source data 1.** Uncropped and labelled gels for *Figure 2—figure supplement 2*.

**Figure supplement 2—source data 2.** Raw unedited gels for *Figure 2—figure supplement 2*.

PLP-dependent amino acid decarboxylases that have been previously characterized (*Komori et al., 2012*; *Huang et al., 2018*).

The crystal structures of the CsAlaDC-EA complex and AtSerDC were obtained and further analyzed. In the former complex, PLP bound to Lys309 via a Schiff base linkage (internal aldimine form), and the pyridine moiety of PLP is positioned between the imidazole ring of His196 and the methyl group of Ala279 in a parallel orientation to the imidazole ring. The carboxylic group of Asp277 stabilizes the N1 of PLP by a salt bridge interaction, providing the latter with the strong electrophilicity necessary to stabilize carbon ion intermediates during enzymatic catalysis. The O3 forms hydrogen bonds with Thr247 and Lys309. Additionally, His308, Gly169, Thr170, Lys309, and Ser347*('*' denotes the amino acids on adjacent subunits) establish a hydrogen bonding network with the phosphate group of PLP, consequently fortifying its attachment to the protein (*Figure 3C*). Similarly, in the latter structure, the pyridine ring of PLP was sandwiched between the imidazole ring of His201 and the methyl group of Ala284. The N1 of PLP formed a salt bridge with Asp282, while the O3 formed hydrogen bonds with Thr252. The phosphate group of PLP established hydrogen bonding interactions with Lys314, His313, Gly174, Thr175, and Ser352* (*Figure 3E*). The aforementioned amino acid residues are observed across various Group II PLP-dependent amino acid decarboxylases and exhibit considerable stability (see further discussion below), suggesting that they could share common features in terms of catalytic mechanisms.

## Factors affecting CsAlaDC and AtSerDC activity

The Group II PLP-dependent amino acid decarboxylases are dimeric enzymes with each monomer containing an active site located within a shallow cavity (*Paiardini et al., 2017*). A flexible loop of amino acids originating from one monomer extends into the active site of the other monomer and plays a crucial role in catalysis. The catalytic loop in AtSerDC, consisting of amino acid residues 328–341, is well-structured and exhibits clear electron density. However, the corresponding loop in CsAlaDC is disordered (*Figure 3F*). The aforementioned loop harbors a conserved Tyr residue that is believed to function as a proton donor of the carbanion in the catalytic process (*Torrens-Spence et al., 2020*). In particular, CsAlaDC Tyr336 and AtSerDC Tyr341 correspond to that Tyr residue (*Figure 3—figure supplement 2*). Upon superimposing the active site residues of CsAlaDC apo and CsAlaDC-EA complex, it was observed that the hydroxyl group of Tyr336* in the CsAlaDC-EA complex exhibits a substantial 60 degree deviation along the Cα-Cβ bond (*Figure 3G*). To assess the function of the Tyr, we substituted the corresponding Tyr residues in CsAlaDC and AtSerDC with Phe. The resulting CsAlaDC^Y336F and AtSerDC^Y341F mutants were then exposed to Ala and Ser, respectively. We measured the production of EA or ethanolamine in the reaction mixtures. Our findings indicate that these mutants catalyze abortive decarboxylation, as evidenced by the absence of detectable EA and only a small amount of

**Table 1.** Kinetic parameters of *Camellia sinensis* alanine decarboxylase (CsAlaDC), *Arabidopsis thaliana* serine decarboxylase (AtSerDC), and *Camellia sinensis* serine decarboxylase (CsSerDC).

| Enzyme | Substrate | $Km$ (mM) | $Vmax$ (µmol L$^{-1}$ s$^{-1}$) | $Kcat$ (s$^{-1}$) | $Kcat/Km$ (s$^{-1}$ M$^{-1}$) |
|---|---|---|---|---|---|
| CsAlaDC | Alanine | 1.215±0.237 | 1.709±0.072 | 0.068 | 56.0 |
| AtSerDC | Serine | 1.912±0.386 | 7.787±0.498 | 0.311 | 162.7 |
| CsSerDC | Serine | 2.364±0.431 | 9.031±0.520 | 0.361 | 152.7 |

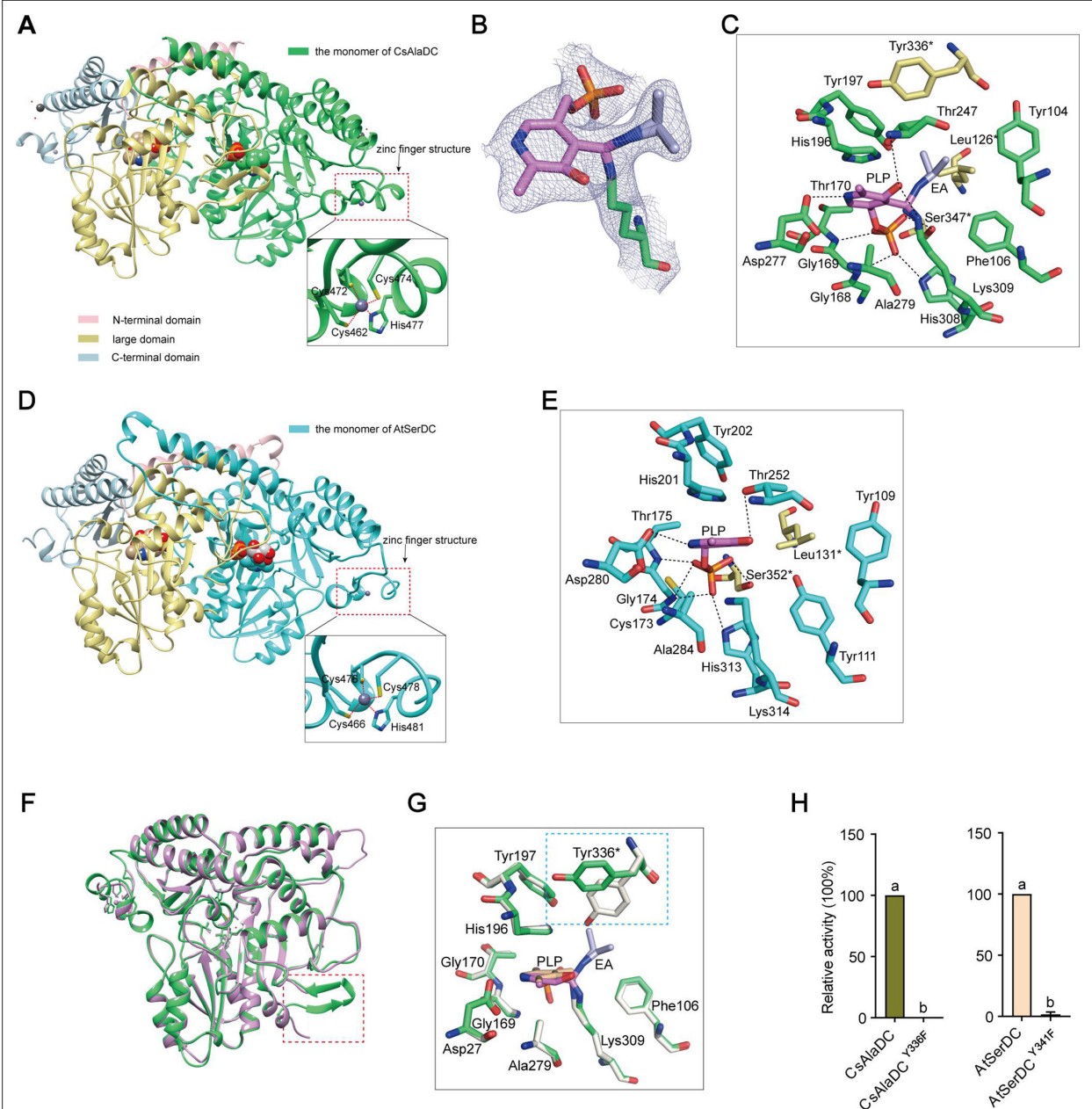

**Figure 3.** Crystal structures of *Camellia sinensis* alanine decarboxylase (CsAlaDC) and *Arabidopsis thaliana* serine decarboxylase (AtSerDC). (**A**) Dimer structure of CsAlaDC. The color display of the N-terminal domain, large domain, and C-terminal domains of chain A is shown in light pink, khaki, and sky blue, respectively. Chain B is shown in spring green. The pyridoxal-5-phosphate (PLP) molecule is shown as a sphere model. The zinc finger structure at the C-terminus of CsAlaDC is indicated by the red box. The gray spheres represent zinc ions, while the red dotted line depicts the coordination bonds formed by zinc ions with cysteine and histidine. (**B**) The 2Fo-Fc electron density maps of K309-PLP-EA (contoured at 1σ level). The PLP is shown in violet, the K309 is shown in spring green, and the ethylamine (EA) is shown in light blue. (**C**) Active center of the CsAlaDC-EA complex, with hydrogen bonds denoted by black dotted lines. '*' denotes the amino acids on adjacent subunits. (**D**) Dimer structure of AtSerDC. The color display of the N-terminal domain, large domain, and C-terminal domains of chain A is shown in light pink, khaki, and sky blue, respectively. Chain B is shown in cyan. The PLP molecule is shown as a sphere model. The zinc finger structure at the C-terminus of AtSerDC is indicated by the red box. The gray spheres represent zinc ions, while the red dotted line depicts the coordination bonds formed by zinc ions with cysteine and histidine. (**E**) Active center of the AtSerDC, with hydrogen bonds denoted by black dotted lines. '*' denotes the amino acids on adjacent subunits. (**F**) The monomers of CsAlaDC and AtSerDC are superimposed. CsAlaDC is depicted in spring green, while AtSerDC is shown in plum. The conserved amino acid catalytic loop is indicated by the red box. (**G**) Amino acid residues of the active center in the CsAlaDC apo and CsAlaDC-EA complex are superimposed. CsAlaDC apo is shown in floral white, while CsAlaDC-EA complex is shown in spring green. (**H**) The relative activity of wild-type CsAlaDC and its Y336F mutant (left), as well as wild-type

*Figure 3 continued on next page*

Figure 3 continued

AtSerDC and its Y341F mutant (right), is shown. Data represent mean ± SD (n=3). The significance of the difference (p<0.05) was labeled with different letters according to Duncan's multiple range test.

The online version of this article includes the following source data and figure supplement(s) for figure 3:

**Source data 1.** Raw excel data for *Figure 3H*.

**Figure supplement 1.** Crystal structures of *Camellia sinensis* alanine decarboxylase (CsAlaDC) and *Arabidopsis thaliana* serine decarboxylase (AtSerDC).

**Figure supplement 2.** Catalytic mechanisms and conformational changes of Camellia sinensis alanine decarboxylase (CsAlaDC).

**Figure supplement 3.** Structures of HisDC2, MetDC, TrpDC, AspDC, HisDC1, and GluDC.

**Figure supplement 4.** Multiple sequence alignments of *Camellia sinensis* alanine decarboxylase (CsAlaDC), *Arabidopsis thaliana* serine decarboxylase (AtSerDC), MetDC, HisDC1, TrpDC, HisDC2, TyrDC, and GluDC were generated using MUSCLE and visualized with ESPript 3 .x.

ethanolamine observed in the reaction mixture (*Figure 3H*). This result suggested this Tyr is required for the catalytic activity of CsAlaDC and AtSerDC.

## Identification of key amino acids for the substrate specificity

The superposition of amino acid residues in the substrate binding pocket of CsAlaDC and AtSerDC revealed that all residues are identical, except for the substitution of Tyr at position 111 in AtSerDC with Phe at position 106 in CsAlaDC (*Figure 4A*). This observation suggests a potential role for this specific residue in determining the selective binding of the appropriate substrate.

To gain further insights into the functional role of this residue (Tyr$^{111}$ in AtSerDC or Phe$^{106}$ in CsAlaDC) in selective binding of the appropriate substrate, we identified 563 potential serine decarboxylases in Embryophyta using the amino acid sequences of AtSerDC (*Figure 4—figure supplement 1A*). By comparing the amino acid sequences of these serine decarboxylases, we observed that the corresponding residues to Tyr$^{111}$ or Phe$^{106}$ are located within a conserved motif comprising 9 amino acids (*Figure 4B*). Within this motif, the first two residues, Y (Tyrosine) and P (Proline) are completely conserved across all 563 proteins. However, the third residue where Tyr$^{111}$ in AtSerDC or Phe$^{106}$ in CsAlaDC is positioned exhibits variability among Y, T (Threonine), F (Phenylalanine), V (Valine), A (Alanine), I (Isoleucine), L (Leucine), or other amino acids (*Figure 4B*). Remarkably similar to Tyr$^{111}$, this residue is predominantly Y in 83.7% of these homologs; conversely, F was found to be present in only 2.3% of these homologs within plant species belonging to Poales, Asterales, Ranunculales, Ericales, Caryophyllales, and Nymphaeales orders (*Figure 4C, D*).

To verify the role of Tyr$^{111}$ in AtSerDC or Phe$^{106}$ in CsAlaDC in the selective binding of the appropriate substrate, we introduced mutations in Tyr$^{111}$ to Phe (AtSerDC$^{Y111F}$) and Phe$^{106}$ to Tyr (CsAlaDC$^{F106Y}$), followed by *in vitro* enzymatic activity assays. The results revealed that AtSerDC$^{Y111F}$ acquired alanine decarboxylase activity, whereas CsAlaDC$^{F106Y}$ completely lost its alanine decarboxylase activity (*Figure 4E*). Additionally, we generated other mutations of Tyr$^{111}$ in AtSerDC, including AtSerDC$^{Y111A}$, AtSerDC$^{Y111I}$, AtSerDC$^{Y111L}$, AtSerDC$^{Y111V}$, and AtSerDC$^{Y111W}$. Among these mutants, only AtSerDC$^{Y111I}$ exhibited a marginal level of alanine decarboxylase activity (*Figure 4E*). These findings indicate the essential role of Phe$^{106}$ in the selective binding of alanine for CsAlaDC.

On the other side, both CsAlaDC and CsAlaDC$^{F106Y}$ exhibit a very low level of serine decarboxylase activity (*Figure 4F*). Moreover, compared with AtSerDC, the serine decarboxylase activity of AtSerDC$^{Y111F}$ was about 30% of AtSerDC; AtSerDC$^{Y111I}$ retained lower than 5% of serine decarboxylase activity of AtSerDC; while other mutations, including AtSerDC$^{Y111A}$, AtSerDC$^{Y111L}$, AtSerDC$^{Y111V}$, and AtSerDC$^{Y111W}$ abolished the serine decarboxylase activity (*Figure 4F*). These results suggested the Tyr$^{111}$ of AtSerDC is also important for the selective binding of serine for AtSerDC.

To further verify that Phe$^{106}$ of CsAlaDC and Tyr$^{111}$ of AtSerDC were key amino acid residues determining the selective binding of substrates *in planta*, we employed the *Nicotiana benthamiana* transient expression system. To this end, *A. tumefaciens* strain GV3101 (pSoup-p19), carrying recombinant plasmid, was infiltrated into leaves of 5-week-old *N. benthamiana* plants, and the pCAMBIA1305 empty vector was used as the control (EV). Relative mRNA levels were detected (*Figure 4—figure supplement 2*), indicating that they had been successfully overexpressed in tobacco. Here, we found that a high level of EA was produced in the CsAlaDC-expressing tobacco leaves; while no EA product was detected in tobacco leaves infiltrated with the mutant CsAlaDC$^{F106Y}$. In addition, we did not detect

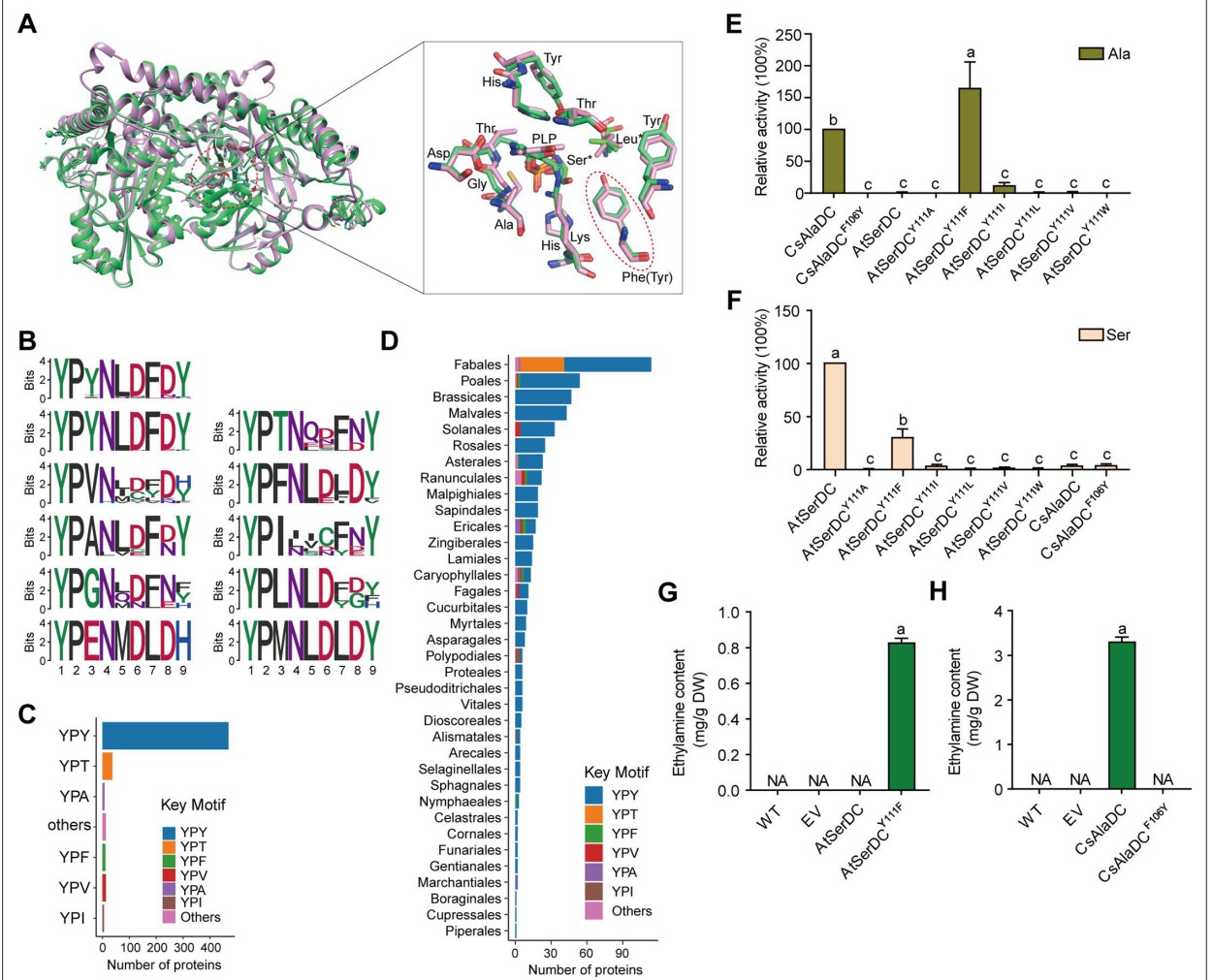

**Figure 4.** Key amino acid residues for substrate recognition. (**A**) Superposition of substrate binding pocket amino acid residues in *Camellia sinensis* alanine decarboxylase (CsAlaDC) and *Arabidopsis thaliana* serine decarboxylase (AtSerDC). The amino acid residues of CsAlaDC are shown in spring green, and the amino acid residues of AtSerDC are shown in plum, with the substrate specificity-related amino acid residue highlighted in a red ellipse. (**B**) Active-site-lining amino acid residues of serine decarboxylase (SDC) homologs from Embryophyta were identified. The height of each amino acid is scaled proportionally to the amount of information content (measured in bits). The first line depicts the conserved motif in all serine decarboxylase (SerDC) homologs from Embryophyta, whereas lines 2–5 represent the conserved motifs based on the variable third amino acid residue. (**C**) Histogram showing the distribution of the number of key motifs. (**D**) Histogram showing the number of key motifs in different plant orders. (**E**) Relative enzyme activities of wild-type CsAlaDC and mutant protein CsAlaDC^F106Y against Ala substrate (columns 1 and 2), and enzyme activities of wild-type AtSerDC and various AtSerDC mutant proteins against Ala substrate (columns 3–9) are presented. The percentage graph shows the relative activity of each protein compared to wild-type CsAlaDC activity (taken as a 100% benchmark). (**F**) Relative enzyme activities of wild-type AtSerDC and AtSerDC mutant proteins (columns 1–7) against Ser substrates, and enzyme activities of wild-type CsAlaDC and mutant protein CsAlaDC^F106Y against Ser substrates (columns 8, 9) were measured. The percentage graph shows the relative activity of each protein compared to the wild-type AtSerDC (taken as a 100% benchmark). Three independent experiments were conducted. (**G**) The EA contents of AtSerDC and its mutant AtSerDC^Y111F in *N. benthamiana*. (**H**) The ethylamine (EA) contents of CsAlaDC and its mutant CsAlaDC^F106Y in *N. benthamiana*. The significance of the difference ($p<0.05$) was labeled with different letters according to Duncan's multiple range test.

The online version of this article includes the following source data and figure supplement(s) for figure 4:

**Source data 1.** Raw excel data for *Figure 4E-H*.

**Figure supplement 1.** Evolutionary analysis of *Camellia sinensis* alanine decarboxylase (CsAlaDC) in Embryophyta.

**Figure supplement 2.** Relative mRNA expression levels of different proteins.

**Figure supplement 3.** Circular Dichroism Spectra of proteins.

**Figure supplement 4.** Absorption Spectra of different proteins.

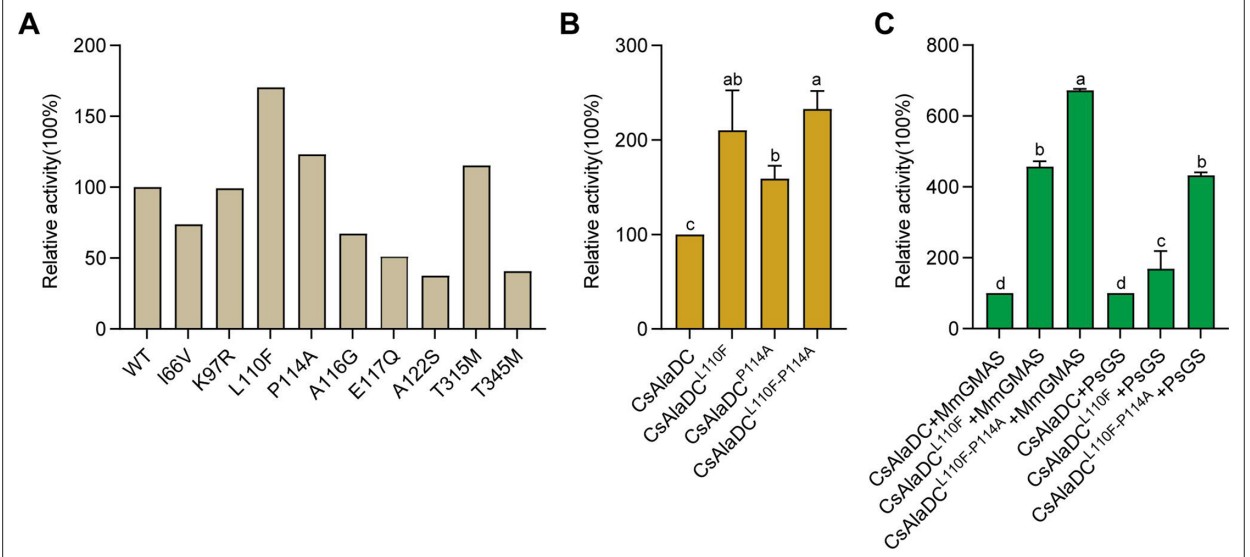

**Figure 5.** Mutations enhance *Camellia sinensis* alanine decarboxylase (CsAlaDC) enzyme activity and theanine synthesis *in vitro*. (**A**) Relative enzyme activities of CsAlaDC mutant proteins against Ala substrate. (**B**) Relative enzyme activities of CsAlaDC[L110F], CsAlaDC[P114A], and CsAlaDC[L110F/P114A] against Ala substrate. (**C**) Histogram showing the relative content of theanine resulting from different combinations of alanine decarboxylase and theanine synthetase. Data represent mean ± SD (n=3). The significance of the difference (p<0.05) was labeled with different letters according to Duncan's multiple range test.

The online version of this article includes the following source data for figure 5:

**Source data 1.** Raw excel data for *Figure 5*.

EA products in the AtSerDC-expressing tobacco leaves, while a high level of EA was detected in tobacco leaves infiltrated with mutant AtSerDC[Y111F]. As anticipated, EA was not detected in tobacco leaves infiltrated with EV (*Figure 4G and H*). These results further verified the critical role of Phe[106] in the selective binding of alanine for CsAlaDC.

## Key amino acids for the evolution of CsAlaDC enzymatic activity

CsAlaDC and AtSerDC have a high sequence similarity of 74.5% and a nearly identical structure with an RMSD (root mean square deviation) of only 0.77 Å for monomer structures. However, the two enzymes catalyze different amino acid decarboxylation reactions, and AtSerDC exhibits significantly higher $V_{max}$ than CsAlaDC (*Figure 2B and C*). In light of this observation, we postulated a hypothesis: EA, generated via Ala decarboxylation by CsAlaDC, can be toxic and harmful to plants if accumulated excessively. Thus, during the evolution of plant serine decarboxylase into alanine decarboxylase, the enzyme has evolved not just to alter the substrate preference but also to reduce catalytic activity to control EA production within a suitable range. Based on this hypothesis, we suggest that mutating specific amino acids in CsAlaDC to those corresponding amino acids in SerDCs could enhance its activity, and the results could provide insights into the evolution of CsAlaDC enzymatic activity.

Dimerization is essential to AlaDC/SerDC activities because the active site is composed of residues from two monomers. To identify the amino acids repressing the enzymatic activity of CsAlaDC during evolution from SerDC, we analyzed the crystal structures and the amino acids at the dimer interface between CsAlaDC, and AtSerDC. This analysis revealed that the amino acids at the dimer interface at positions 66, 97, 110, 114, 116, 117, 122, 315, and 345 are different in CsAlaDC and SerDCs (*Figure 1B*). Therefore, we mutated these amino acids of CsAlaDC into those in the corresponding positions of AtSerDC and CsSerDC, and performed enzyme activity assays (*Figure 5A*). The results demonstrated that CsAlaDC[L110F] and CsAlaDC[P114A] exhibited significantly enhanced enzyme activity compared to the wild-type CsAlaDC, with a 2.1-fold and 1.59-fold increase, respectively. Furthermore, the catalytic activity of the CsAlaDC[L110F/P114A] double mutant exhibits a remarkable 2.3-fold increase compared to that of the wild-type protein (*Figure 5B*). These findings suggested a critical role of Leu[110] and Pro[114] of CsAlaDC in the evolution of enzymatic activity and provided a basis to improve

CsAlaDC activity. It is possible that these amino acid residues could potentially augment the hydrophobic nature of the protein dimer interface.

## *In vitro* synthesis of theanine

The biosynthetic pathway of theanine in tea plants comprises two consecutive enzymatic steps: alanine decarboxylase facilitates the decarboxylation of alanine to generate EA, while theanine synthetase catalyzes the condensation reaction between EA and Glu to synthesize theanine. By conducting mutation screens, we discovered a CsAlaDC mutant protein (L110F/P114A) that exhibited a 2.3-fold higher catalytic activity compared to the wild-type protein. Subsequently, we employed an *in vitro* theanine synthesis system utilizing CsAlaDC and either glutamine synthetase (PsGS [P*seudomonas syringae* pv. syringae]) or gamma-glutamate methylamine ligase (MmGMAS [*Methylovorus mays*]) which have the ability to synthesize theanine from EA and Glu (*Hagihara et al., 2021*; *Yamamoto et al., 2007*).

The results illustrated the successful synthesis of theanine using CsAlaDC in conjunction with two theanine synthetases, employing Ala and Glu as substrates (*Figure 5C*). The theanine content generated by the combination of CsAlaDC$^{L110F}$ and MmGMAS, as well as CsAlaDC$^{L110F/P114A}$ and MmGMAS, was 4.57-fold and 6.72-fold higher than the content produced in the combination of wild-type CsAlaDC and MmGMAS, respectively. Similarly, when combined with the PsGS, comparable outcomes are observed as well. The theanine content resulting from the combination of CsAlaDC$^{L110F}$ and PsGS, as well as CsAlaDC$^{L110F/P114A}$ and PsGS, exhibit enhancements of 1.62-fold and 4.33-fold compared to the wild-type protein combination, respectively (*Figure 5C*). Hence, the utilization of CsAlaDC$^{L110F/P114A}$ could effectively enhance the theanine production yield and thus, holds potential for large-scale engineering production of theanine.

## Discussion

### The common and distinctive structural characteristics of CsAlaDC and AtSerDC compared with other amino acid decarboxylases

CsAlaDC and AtSerDC are Group II PLP-dependent amino acid decarboxylases, which exhibit numerous structural features common to other amino acid decarboxylases. The Dali search (*Holm, 2022*) revealed that AtSerDC and CsAlaDC share structural similarities with 7CIG (MetDC [*Streptomyces sp.* 590]), 7ERV (HisDC1 [*Photobacterium phosphoreum*]), 6KHO (TrpDC [*Oryza sativa Japonica* Group]), 4E1O (HisDC2 [*Homo sapiens*]), 6JY1 (AspDC [*Methanocaldococcus jannaschii*]), and 5GP4 (GluDC [*Levilactobacillus brevis*]). The monomers of these amino acid decarboxylases all consist of three characteristic domains: the C-terminal domain, the large domain, and the N-terminal domain (*Figure 3—figure supplement 3*). The amino acid residues that bind to PLP cofactors are conserved across multiple enzymes (*Figure 3—figure supplement 3* and *Figure 3—figure supplement 4*). Out of the eight proteins analyzed, a total of thirteen amino acid residues were found to be conserved across all sequences (Glu$^{143}$, Glu$^{171}$, Lys$^{201}$, Gly$^{245}$, Thr$^{247}$, Asp$^{253}$, His$^{275}$, Asp$^{277}$, Ala$^{279}$, Pro$^{286}$, Ser$^{302}$, Lys$^{309}$, and Tyr$^{336}$ in CsAlaDC). Some of them are situated within the active center and play a role in stabilizing PLP or facilitating catalytic reactions. Conversely, other residues reside outside the active center and their function remains unclear (*Figure 3—figure supplement 4*).

Group II PLP-dependent amino acid decarboxylases are characterized by the existence of a highly flexible loop, which is of great significance for the catalytic mechanism of decarboxylase (*Okawa et al., 2021*). The loops are located at the dimer interface and extend to the active sites of other monomers in a closed conformation. Prior research has established that the conserved amino acid residue Tyr within the loop plays a crucial role in catalysis by donating protons to carbanions of quinone intermediates that arise following decarboxylation (*Fenalti et al., 2007*; *Chellam Gayathri and Manoj, 2019*). Our experiments demonstrate that the substitution of the corresponding Tyr with Phe in the loop renders the protein inactive. Through the application of circular dichroism, we have corroborated that the stability of the protein is not compromised by mutations (*Figure 4—figure supplement 3A, B, D, E*). Both the mutant and wild-type proteins manifest absorption bands at the 420 nm wavelength, signifying the formation of a Schiff base between PLP and the lysine residues found at the active site. Given that PLP has been incorporated during the protein purification stage, the absorbance observed for both the mutant and wild-type proteins at 420 nm is consistent when compared at equivalent concentrations (*Figure 4—figure supplement 4*). This corroboration implies that the protein mutation

does not interfere with the binding to PLP. Consequently, the inactivation observed in the mutant proteins is not a consequence of instability incited by corresponding mutations, nor changes in the binding affinity between the protein and PLP induced by these mutations.

We observed that the substitution of Phe[106] for Tyr in CsAlaDC rendered CsAlaDC inactive, while the substitution of Tyr[111] for Phe in AtSerDC enabled alanine decarboxylase activity, and the alteration in the activity of the mutant protein bears no correlation to protein stability and affinity for PLP (*Figure 4—figure supplement 3A, C, D, F* and *Figure 4—figure supplement 4*), suggesting that the amino acid in the position is critical for determining substrate specificity. Integrative crystallographic structure of the two enzymes, we speculate the hydrophilic nature of the Lys residue in AtSerDC predisposes the active site to preferentially accommodate the hydrophilic amino acid Ser as its substrate. In contrast, the equivalent position in CsAlaDC is occupied by Phe, an amino acid lacking the hydroxyl group. This substitution enhances the hydrophobic nature of the substrate-binding pocket. Consequently, CsAlaDC demonstrates a unique predilection, selectively binding Ala (an amino acid with comparatively hydrophobic properties) as its preferred substrate.

The monomeric configuration of CsAlaDC and AtSerDC is akin to that of other Group II PLP-dependent amino acid decarboxylases, except for a conspicuous dissimilarity. Unlike other Group II PLP-dependent amino acid decarboxylases, the C-terminal of CsAlaDC and AtSerDC harbors an evident zinc finger structure. We propose that this structure could potentially influence enzyme stability. To test this hypothesis, we truncated the zinc finger structure from both proteins and expressed them. Our findings indicate that, following the excision of the zinc finger structure, CsAlaDC became insoluble, while AtSerDC displayed a similar trait to a certain extent (*Figure 2—figure supplement 2*). Hence, we can establish that the zinc finger structure indeed exerts an influence on the enzyme's stability. The potential for additional functions necessitates further investigation.

## Evolution of AlaDC and SerDC

The structure of CsAlaDC seems like with AtSerDC, however, there are differences in substrate specificity. To clarify the relationship between CsAlaDC and other SerDCs, we analyzed serine decarboxylase-like proteins in Embryophyta that are highly homologous to CsAlaDC and constructed phylogenetic trees to gain insight into their evolutionary relationships (*Figure 4—figure supplement 1*). Based on our experimental findings and evolutionary evidence, we have identified a conserved YPX motif at the substrate binding pocket. For most plants, the residue X in this motif is predominantly Tyr, which corresponds to the serine decarboxylase protein. However, other residues such as Thr, Phe, Val, Ala, and Ile can also occupy this position (*Figure 4B and C*). Interestingly, serine decarboxylase-like proteins containing YPF motifs, which have the potential to function as alanine decarboxylases, were found to be distributed throughout the phylogenetic tree, including Asteroideae, Ericales, Chenopodiaceae, Poaceae, and Ranunculales (*Figure 4—figure supplement 1B*). However, these proteins were absent in some more recent species. This observation implies that the emergence of alanine decarboxylase in these particular species could be attributed to convergent evolution.

Moreover, we have observed the enrichment of serine decarboxylase-like proteins that possess a YPT motif within the Fabales (*Figure 4—figure supplement 1B*). Prior research has demonstrated that XP_004496485–1, which possesses a YPT motif, displays catalytic efficacy toward the processes of decarboxylation and oxidative deamination of Phe, Met, Leu, and Trp (*Torrens-Spence et al., 2014*). Given that the YPT motif is highly conserved and widely distributed in Fabales, serine decarboxylase-like proteins bearing the YPT motif may have developed a unique substrate specificity in Fabales, beyond their conventional decarboxylation functions, as exemplified by XP_004496485–1 protein mentioned above. We speculate that these proteins may be capable of catalyzing other reactions, potentially involving non-protein amino acids or other substances as substrates.

## Applied to improve the synthesis of theanine

Theanine is an important indicator of green tea quality. Therefore, improving the synthesis of theanine in tea plants is the focus of research. In this study, through crystal structure analysis and mutation verification, we found that the catalytic activity of CsAlaDC[L110F/P114A] is 2.3 times higher than that of the wild-type protein, resulting in a more abundant synthesis of theanine *in vitro*. This gives us great inspiration to improve the synthesis of EA in tea plants by gene editing, thus increasing the content of theanine in tea plants (*Figure 5*).

Theanine is highly demanded, by the market, due to its health effects and medicinal value, and as a constituent in food, cosmetics, and other fields. To meet the market demand, a variety of methods have been used to acquire theanine, with the main methods including direct extraction, chemical synthesis, biotransformation (microbial fermentation), and plant cell culture. Based on the findings of this study, site-directed mutagenesis can be employed to modify enzymes involved in theanine synthesis. This modification enhances the capacity of bacteria, yeast, model plants, and other organisms to synthesize theanine, thereby facilitating its application in industrial theanine production.

## Conclusions

In conclusion, our structural and functional analyses have significantly advanced understanding of the substrate-specific activities of alanine and serine decarboxylases, typified by CsAlaDC and AtSerDC. Critical amino acid residues responsible for substrate selection were identified-Tyr111 in AtSerDC and Phe106 in CsAlaDC-highlighting pivotal roles in enzyme specificity. The engineered CsAlaDC mutant (L110F/P114A) not only displayed enhanced catalytic efficiency but also substantially improved L-theanine yield in a synthetic biosynthesis setup with PsGS or MmGMAS. Our research expanded the repertoire of potential alanine decarboxylases through the discovery of 13 homologous enzyme candidates across embryophytic species and uncovered a special motif present in serine protease-like proteins within Fabale, suggesting a potential divergence in substrate specificity and catalytic functions. These insights lay the groundwork for the development of industrial biocatalytic processes, promising to elevate the production of L-theanine and support innovation within the tea industry.

## Materials and methods

### Plants materials

Tobacco (*Nicotiana benthamiana*) plants were grown in a controlled chamber, under a 16 hr light and 8 hr dark photoperiod at 25°C. Leaves of 5-week-old tobacco plants were used for transient transformation, mediated by *Agrobacterium tumefaciens* strain GV3101.

### Gene cloning and protein expression

The coding sequences of AtSerDC and CsAlaDC, amino acid residues 66–482 and 61–478, respectively, were amplified using PCR and then ligated into the *Nde* Ⅰ and *Xho* I restriction sites of the pET-28a and pET-22b vector, respectively, providing the recombinant vector pET-28a-*AtSerDC* and pET-22b-*CsAlaDC*. The cDNAs encoding CsSerDC were amplified using PCR and then ligated into the *Nde* Ⅰ and *Xho* I restriction sites of the pET-28a vector. Gene-specific primers were listed in *Supplementary file 1b*. The recombinant plasmid was transformed into *E. coli* BL21 (DE3) competent cells. Positive transformants were grown in a 5 mL LB medium containing 30 μg/mL kanamycin or 50 μg/mL ampicillin at 37 °C overnight and then subcultured into an 800 mL LB medium containing the corresponding antibiotic. Protein expression was induced by the addition of 0.2 mM isopropyl-β-d-thiogalactoside (IPTG) for 20 hr at 16°C when the optical density (OD) at 600 nm reached 0.6–0.8, harvested by centrifugation at 4°C and 4000 rpm for 30 min.

### Protein production and crystallization

The cell pellet was suspended in 30 mL of lysis buffer (20 mM HEPES, pH 7.5, 200 mM NaCl, 0.1 mM PLP), disrupted by a High-Pressure Homogenizer, and then centrifuged at 16,000 rpm for 30 min at 4 °C to remove the cell debris. The supernatants were purified with a Ni-Agarose resin column followed by size-exclusion chromatography. Before crystallization, purified proteins were concentrated at 10 mg/mL. Crystallization conditions were screened by the sitting-drop vapor diffusion method using the reservoir solutions supplied in commercially available screening kits (Crystal Screen, Crystal Screen 2, PEGRx 1, 2, and SaltRx 1, 2). A droplet made by mixing 1.0 μL of purified AtSerDC or CsAlaDC (10 mg/mL) with an equal volume of a reservoir solution was equilibrated against 100 μL of the reservoir solution at 16°C. The crystal of AtSerDC was obtained using buffer pH 8.0 containing 20% (w/v) PEG400 as a precipitate and 0.2 M CaCl$_2$. The crystal of the CsAlaDC-EA complex was obtained at pH 7.5 containing 2.6 M sodium acetate. The crystal of CsAlaDC was obtained at pH 6.0 containing 3.5 M sodium formate.

## Data collection and processing

Crystals were grown by sitting-drop vapor diffusion method at 16 °C. The volume of the reservoir solution was 100 μL and the drop volume was 2 μL, containing 1 μL of protein sample and 1 μL of reservoir solution. The reservoir solution of AtSerDC contained 0.1 M Tris-HCl (pH 8.0), 0.2 M $CaCl_2$, and 20% PEG 400. The reservoir solution of CsAlaDC contained 0.1 M HEPES (pH 7.5), 2.6 M sodium acetate or 0.1 M Bis-Tris (pH 6.0), and 3.5 M sodium formate. Crystals grew in 3–5 days using a protein concentration of 10 mg/mL. Diffraction data were collected at the Shanghai Synchrotron Radiation Facility (China). The collected data sets were indexed, integrated, and scaled using the HKL3000 software package. The structure of AtSerDC was solved by molecular replacement using the structure of HisDC (PDB code: 7ERV [*Photobacterium phosphoreum*]) as the model, and utilizing AtSerDC as a molecular displacement model for both CsAlaDC and the CsAlaDC-EA complex. The AtSerDC, CsAlaDC, and CsAlaDC-EA complex exhibit resolutions of 2.85 Å, 2.50 Å, and 2.60 Å, respectively. The statistics for data collection and processing are summarized in *Supplementary file 1a* .

## Enzyme activity assays

Decarboxylase activity was measured by detecting products (EA or ethanolamine) in the Waters Acquit ultraperformance liquid chromatography (UPLC) system (*Bai et al., 2021*; *Qu et al., 2021*). The 100 μL reaction mixture, containing 20 mM substrate (Ala or Ser), 100 mM potassium phosphate, 0.1 mM PLP, and 0.025 mM purified enzyme, was prepared and incubated at standard conditions (45 °C and pH 8.0 for CsAlaDC, 40 °C and pH 8.0 for AtSerDC for 10 min). Then, the reaction was stopped with 20 μL of 10% trichloroacetic acid. The product was derivatized with 6-aminoquinolyl-N-hydroxy-succinimidyl carbamate (AQC) and subjected to analysis by UPLC. All enzymatic assays were performed in triplicate.

The detection methodology for theanine production remains consistent with the aforementioned approach. The 100 μL reaction mixture, containing 20 mM Ala, 45 mM Glu, 50 mM HEPES (pH 7.5), 0.1 mM PLP, 30 mM $MgCl_2$, 10 mM ATP, 0.03 mM PsGS/MmGMAS, and 0.025 mM CsAlaDC/CsAlaDC[L110F]/CsAlaDC[L110F/P114A], was prepared and incubated at standard conditions for 1 h. Subsequently, the reaction was terminated via immersion of the reaction vessel in a metal bath at 96 °C for 3 min (*Yamamoto et al., 2007*). Theanine was derivatized with AQC and subjected to analysis by UPLC. All enzymatic assays were performed in triplicate.

## Site-directed mutagenesis

Site-directed mutagenesis experiment was conducted using a PCR method from the wild-type construct pET-28a-*AtSerDC* and pET-22b-*CsAlaDC*, respectively. *Dpn* I endonucleases were used to digest the parental DNA template. The reaction mixture was used to transform *E. coli* DH5α competent cells and the plasmids from positive strains were extracted to *E. coli* BL21 (DE3) for protein expression, purification, and analysis of the enzymatic activity.

## *In vivo* enzyme activity assay in *N. benthamiana*

The amplified PCR products were fused to the plant expression vector, pCAMBIA1305. Linearization was conducted by restriction digest with *Spe* I and *Bam*H I. The recombinant colonies were selected for PCR validation on the appropriate antibiotics plate. After validation, the plasmids were electroporated into *Agrobacterium tumefaciens* strain GV3101(pSoup-p19). Empty pCAMBIA1305 vector, with an intron-containing the GFP gene, was treated as the control.

*Agrobacterium* transient expression assays were performed on 5-week-old *N. benthamiana* plants. *Agrobacterium tumefaciens* strain GV3101 (pSoup-p19), carrying the above-described vectors, were cultured in Luria Bertani (LB) medium, containing appropriate antibiotics, at 28 °C. When the absorbance of bacteria colonies reached $OD_{600}$=0.6–0.8, bacterial cells were collected and resuspended in MMA solution (10 mM $MgCl_2$, 10 mM 2-(N-morpholino) ethane sulfonic acid (MES), pH 5.6). After the $OD_{600}$ of the resuspended bacterial solution was adjusted to approx. 1.0, acetosyringone (AS) was added with a final concentration of 200 μM, and this solution was then incubated, at room temperature, for at least 3 hr in darkness. Next, cell suspensions were infiltrated into *N. benthamiana* leaves with a needle-free syringe. These *N. benthamiana* leaves were then collected 3 days post infiltration, frozen in liquid nitrogen, and stored at –80 °C. Internal EA in *N. benthamiana* leaves was

extracted as previously described (*Berger et al., 2009*), and then the solvent was subjected to a gas chromatography-mass spectrometry GS-MS system.

## Transcript level analysis in *N. benthamiana*

Total RNA was isolated from samples using the RNAprep Pure Plant Kit (Tiangen, Beijing, China), according to the manufacturer's protocol. The cDNAs were synthesized using TransScript One-Step gDNA Removal and cDNA Synthesis SuperMix Kit (TransGen Biotech, Beijing, China). The qRT-PCR assays were performed, as previously described (*Madeira et al., 2022*). Primers used for qRT-PCR assays were listed in *Supplementary file 1c* and the qRT-PCR was run on a Bio-Rad CFX96TM RT PCR detection system and CFX Manager Software. Each reaction reagent (20 µL) contained 0.4 µL forward and reverse primers (10 µM), 2 µL cDNA (200±5 ng/µL), 10 µL SYBR Green Supermix (Vazyme, Nanjing, China), and 6.2 µL double-distilled water. Reaction was performed by a two-step method: 95°C for 5 min; 40 cycles of 95°C for 10 s; and 60°C for 30 s. The glyceraldehyde-3-phosphate dehydrogenase (GAPDH) gene was used for internal normalization in each RT-PCR, and the $2^{-\Delta\Delta CT}$ method (*Dong et al., 2020*) was used to calculate the relative gene expression. All samples were performed with three replicates.

## Multiple sequence alignment

MUSCLE was used to generate the protein multiple sequence alignment of amino acid decarboxylases with default settings (*Schmittgen et al., 2000*). ESPript 3 .x was used to display the multiple sequence alignment (*Robert and Gouet, 2014*).

## Phylogenetic analysis

The AtSerDC was used as a query for searching its homologs in Embryophyta using BLASTP (*Camacho et al., 2009*) (version 2.13.0). To fetch homologs of AtSerDC in distant species, we performed a two-round blast. We first search for the best hit in the given order under Embryophyta. Then we blast each species under a provided order using the order best hit as a query and as many as 10 hits were kept. The sequences with e values ≤ 0.001 were removed. We further dropped sequences with lengths less than 200 or larger than 800 amino acids. Then multiple sequence alignment was performed via clustalo (version 1.2.4) with default settings on these filtered sequences (*Sievers and Higgins, 2018*). The NJ algorithm implemented in clustalw (version 2.1) was adopted to build the phylogenetic tree (*Larkin et al., 2007*). The R package ggtree (version 3.2.1) and universalmotif (version 1.12.4) were used for tree and motif visualization (*Yu et al., 2018*).

## Acknowledgements

We thank the team of beamline BL18U1 in the Shanghai Synchrotron Radiation Facility for diffraction data collection.

## Additional information

### Funding

| Funder | Grant reference number | Author |
|---|---|---|
| Ministry of Science and Technology of the People's Republic of China | 2019YFA0904100 | Weimin Gong |
| Ministry of Science and Technology of the People's Republic of China | 2022YFF1003103 | Xiaochun Wan |
| Ministry of Science and Technology of the People's Republic of China | T2221005 | Weimin Gong |

| Funder | Grant reference number | Author |
|---|---|---|
| Ministry of Science and Technology of the People's Republic of China | 32072624 | Zhaoliang Zhang |

The funders had no role in study design, data collection and interpretation, or the decision to submit the work for publication.

## Author contributions

Hao Wang, Data curation, Software, Formal analysis, Validation, Investigation, Visualization, Methodology, Writing - original draft, Writing - review and editing; Biying Zhu, Formal analysis, Validation, Visualization, Methodology, Writing - original draft, Writing - review and editing; Siming Qiao, Chunxia Dong, Xiaochun Wan, Investigation; Weimin Gong, Conceptualization, Resources, Data curation, Software, Supervision, Funding acquisition, Project administration, Writing - review and editing; Zhaoliang Zhang, Conceptualization, Resources, Data curation, Supervision, Funding acquisition, Project administration, Writing - review and editing

## Author ORCIDs

Hao Wang (ID) https://orcid.org/0009-0003-8119-9882
Biying Zhu (ID) https://orcid.org/0009-0000-9652-7787
Weimin Gong (ID) https://orcid.org/0000-0002-6723-1298
Zhaoliang Zhang (ID) https://orcid.org/0000-0001-6615-1598

Reviewer #2 (Public review): https://doi.org/10.7554/eLife.91046.4.sa1
Reviewer #3 (Public review): https://doi.org/10.7554/eLife.91046.4.sa2
Author response https://doi.org/10.7554/eLife.91046.4.sa3

# Additional files

## Supplementary files

• MDAR checklist

• Supplementary file 1. Supplementary tables. (**a**) Data collection and refinement statistics. (**b**) Primers used for gene cloning. (**c**) Primers used for real-time PCR.

## Data availability

Diffraction data have been deposited in PDB under the accession code 8JG7, 8JIK and 8JIJ respectively. Figure 2—source data 3, Figure 3—source data 1, Figure 4—source data 1, and Figure 5—source data 1 contain the numerical data used to generate the figures.

The following datasets were generated:

| Author(s) | Year | Dataset title | Dataset URL | Database and Identifier |
|---|---|---|---|---|
| Wang H, Gong W | 2024 | Serine decarboxylase | https://www.rcsb.org/structure/8JG7 | RCSB Protein Data Bank, 8JG7 |
| Wang H, Gong W | 2024 | Alanine decarboxylase | https://www.rcsb.org/structure/8JIK | RCSB Protein Data Bank, 8JIK |
| Wang H, Gong W | 2024 | Alanine decarboxylase | https://www.rcsb.org/structure/8JIJ | RCSB Protein Data Bank, 8JIJ |

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
