## [Editor Report · eLife assessment]

This study reports a comparative biochemical and structural analysis of two PLP decarboxylase enzymes from plants. The work is **useful** because of the potential application of these enzymes in industrial theanine production. The structure provides a **solid** basis for understanding substrate specificity but some aspects of the work are **incomplete**. The paper will be of interest to enzymologists studying PLP enzymes and those working on enzyme engineering in plants.

---

## [Referee Report · Reviewer #2 (Public review)]

Summary:

The manuscript focuses on comparison of two PLP-dependent enzyme classes that perform amino acyl decarboxylations. The goal of the work is to understand the substrate specificity and factors that influence catalytic rate in an enzyme linked to theanine production in tea plants.

Strengths:

The work includes x-ray crystal structures of modest resolution of the enzymes of interest. These structures provide the basis for design of mutagenesis experiments to test hypotheses about substrate specificity and the factors that control catalytic rate. These ideas are tested via mutagenesis and activity assays, in some cases both in vitro and in plants.

Weaknesses:

Although improved in a revision, the manuscript could be more clear in explaining the contents of the x-ray structures and how the complexes studied relate to the reactant and product complexes. Some of the figures lack sufficient clarity and description. Some of the claims about the health benefits of tea are not well supported by literature citations.

---

## [Referee Report · Reviewer #3 (Public review)]

In the manuscript titled "Structure and Evolution of Alanine/Serine Decarboxylases and the Engineering of Theanine Production," Wang et al. solved and compared the crystal structures of Alanine Decarboxylase (AlaDC) from *Camellia sinensis* and Serine Decarboxylase (SerDC) from *Arabidopsis thaliana*. Based on this structural information, the authors conducted both in vitro and in vivo functional studies to compare enzyme activities using site-directed mutagenesis and subsequent evolutionary analyses. This research has the potential to enhance our understanding of amino acid decarboxylase evolution and the biosynthetic pathway of the plant specialized metabolite theanine, as well as to further its potential applications in the tea industry.

---

## [Author Response]

The following is the authors’ response to the previous reviews.

Response to reviewer’s comments

**Reviewer #2 (Public Review):**
Summary:The manuscript focuses on comparison of two PLP-dependent enzyme classes that perform amino acyl decarboxylations. The goal of the work is to understand the substrate specificity and factors that influence catalytic rate in an enzyme linked to theanine production in tea plants.Strengths:The work includes x-ray crystal structures of modest resolution of the enzymes of interest. These structures provide the basis for design of mutagenesis experiments to test hypotheses about substrate specificity and the factors that control catalytic rate. These ideas are tested via mutagenesis and activity assays, in some cases both in vitro and in plants.Weaknesses:Although improved in a revision, the manuscript could be more clear in explaining the contents of the x-ray structures and how the complexes studied relate to the reactant and product complexes. The manuscript could also be more concise, with a discussion section that is largely redundant with the results and lacking in providing scholarly context from the literature to help the reader understand how the current findings fit in with work to characterize other PLP-dependent enzymes or protein engineering efforts. Some of the figures lack sufficient clarity and description. Some of the claims about the health benefits of tea are not well supported by literature citations.

Thank you for your insightful comments on our manuscript and your recognition of the strengths of our study. We understand your concerns about the weaknesses mentioned, and we have addressed them appropriately in the revised manuscript. We acknowledge that the discussion section needs to be improved for conciseness and context. We have revised this part by removing the redundant content. We also acknowledge your comments concerning the clarity and description of some figures. We have revisited these figures and revised them, ensuring they are clear and adequately described. Lastly, concerning the claims about the health benefits of tea, we understand your concern about the lack of supporting citations. We ensure to back such claims with valid literature or, if necessary, omit these statements.

**Reviewer #2 (Recommendations For The Authors):**
(1) Line 21: Alanine Decarboxylase should not be capitalized.

Thank you very much for your careful reading of the manuscript. We have corrected it in the revised manuscript.

(2) Line 31: Grammatical error. Also not clear what "evolution analysis" means here. Revise to "Structural comparisons led us to..."

Thank you very much for your careful reading of the manuscript. We have corrected it in the revised manuscript.

(3) Line 34: Revise to "Combining a double mutant of CsAlaDC"

Thank you very much for your careful reading of the manuscript. We have corrected it in the revised manuscript.

(4) Line 35: Change word order to "increased theanine production 672%"

Thank you very much for your careful reading of the manuscript. We have corrected it in the revised manuscript.

(5) Line 37: meaning unclear. Revise to "provides a route to more efficient biosynthesis of theanine."

Thank you very much for your careful reading of the manuscript. We have corrected it in the revised manuscript.

(6) Line 44: I'm not sure that the "health effects" of tea have been proven in placebo controlled studies. And the references provided (2-4 and 5) do not describe original research articles supporting these claims. I would suggest removing these statements from the introduction and at later points in the manuscript.

Thank you for your thoughtful feedback and suggestions. Based on your suggestion, we have removed these statements: "The popularity of tea is determined by its favorable flavor and numerous health benefits (2-4). The flavor and health-beneficial effects of tea are conferred by the abundant secondary metabolites, including catechins, caffeine, theanine, volatiles, etc (5). " As for the subsequent statement: " It has also many health-promoting functions, including neuroprotective effects, enhancement of immune functions, and potential anti-obesity capabilities, among others. " the referenced literature cited can substantiate this conclusion.

(7) Line 58: insert "the" between provided and basis

Thank you very much for your careful reading of the manuscript. We have corrected it in the revised manuscript.

(8) Line 100: Not clear what this phrase means, "As expected, CsSerDC was closer to AtSerDC" Please clarify - closer to what?

We apologize for any confusion caused by the unclear phrasing. When referring to "CsSerDC was closer to AtSerDC," we intended to convey that CsSerDC exhibits a higher degree of sequence homology with AtSerDC than it does with the other enzymes evaluated in our investigation. However, a 1.29% difference between 86.21% and 84.92% in amino acid similarity is not statistically significant (Figure 1B and Supplementary table 1 in the original manuscript), we have deleted the relevant descriptions in the revised manuscript.

(9) Line 112: "were constructed into" makes no sense. It would be better to say the genes for the proteins of interest were inserted into the overexpression plasmid.

Thank you very much for your careful reading of the manuscript. We have corrected it in the revised manuscript.

(10) Line 115: missing the word "the" between generated and recombinant

Thank you very much for your careful reading of the manuscript. We have corrected it in the revised manuscript.

(11) Line 121: catalyze not catalyzed

Thank you very much for your careful reading of the manuscript. We have corrected it in the revised manuscript.

(12) Lines 129 and 130: The reported Km values are really large - in the mM range. Do these values make sense in terms of the available concentrations of the substrates inside the cell?

The content of alanine in tea plant roots ranges from 0.28 to 4.18 mg/g DW (Yu et al., 2021; Cheng et al., 2017). Correspondingly, the physiological concentration of alanine is 3.14 mM to 46.92 mM, in tea plant roots. The content of serine in plants ranges from 0.014 to 17.6 mg/g DW (Kumar et al., 2017). Correspondingly, the physiological concentration of serine is 0.13 mM to 167.48 mM in plants. Therefore, in this study, the Km values are within the range of available substrate concentrations inside the cell.

Yu, Y. *et al*. (2021) Glutamine synthetases play a vital role in high accumulation of theanine in tender shoots of albino tea germplasm "Huabai 1". *J. Agric. Food Chem*. 69 (46),13904-13915.

Cheng, S. *et al*. (2017) Studies on the biochemical formation pathway of the amino acid L-theanine in tea (*Camellia sinensis*) and other plants. *J. Agric. Food Chem.* 65 (33), 7210-7216.

Kumar, V. *et al*. (2017) Differential distribution of amino acids in plants. Amino Acids. 49(5), 821-869.

(13) Line 211: it is unclear what the phrase "as opposed to wild-type" means. Please clarify.

Thank you very much for your careful reading of the manuscript and valuable suggestions. We intend to communicate that the wild-type CsAlaDC and AtSerDC demonstrate decarboxylase activity, while the mutated proteins have experienced a loss of decarboxylation activity. We have already modified this concern in the revised version of the manuscript.

(14) Line 222: residues not residue

Thank you very much for your careful reading of the manuscript. We have corrected it in the revised manuscript.

(15) Line 227 and Figure 4B: It is not clear what the different sequence logos mean in this part of the figure. The caption is too brief and not helpful. And the sentences describing this figure panel are also not sufficiently clear.

Thank you very much for your careful reading of the manuscript and valuable suggestions. We have provided a more detailed explanation of this section in the revised manuscript and added additional annotations in the figure caption to provide further clarity.

(16) Lines 233 and 234: "in the substrate specificity" is awkwardly worded. I would revise to "in selective binding of the appropriate substrate."

Thank you very much for your careful reading of the manuscript and valuable suggestions. We have meticulously revised the description of this section.

(17) Line 243: a word is missing in this sentence - but I can't figure out the intended meaning or what the missing word is. Rephrase to improve clarity.

Thank you very much for your careful reading of the manuscript and valuable suggestions. We have revised this sentence to: " These findings indicate the essential role of Phe106 in the selective binding of alanine for CsAlaDC. "

(18) Line 255: The "expression system...was carried out" is not correct. I would say the expression system was used - but you probably also want to rearrange the sentences to more directly say what it was used for. Later, the word "the" is also missing.

Thank you very much for your careful reading of the manuscript and valuable suggestions. We have revised this sentence to: "To further verify that Phe106 of CsAlaDC and Tyr111 of AtSerDC were key amino acid residues determining its substrate recognition in planta, we employed the *Nicotiana benthamiana* transient expression system. "

(19) Line 273: use "understand" instead of "elucidate" and instead of "we proposed a prediction test:" say "we designed a test of the prediction that..."

Thank you very much for your careful reading of the manuscript. We have revised this sentence to: “In light of this observation, we postulated a hypothesis:”

(20) Line 301: I don't think "effectuate" is a word. Replace with something else.

Thank you very much for your careful reading of the manuscript. We have revised the sentence as: " The biosynthetic pathway of theanine in tea plants comprises two consecutive enzymatic steps: alanine decarboxylase facilitates the decarboxylation of alanine to generate EA, while theanine synthetase catalyzes the condensation reaction between EA and Glu to synthesize theanine. "

(21) Line 307: replace "activity" with "ability"

Thank you very much for your careful reading of the manuscript. We have corrected it in the revised manuscript.

(22) Line 322: I didn't find the discussion very useful. Much of it is simply a recap of the results - which is not necessary. The structural comparisons are overly descriptive without providing appropriate rationale or topic sentence structure so that the reader understands why certain details are emphasized. I think the manuscript would be much stronger if this section were not included or integreted more concisely into the results section where appropriate.

Thank you for your constructive comments. We understand your concerns about the discussion section of our manuscript. We acknowledge that the discussion section has redundancies with the result. In response to this, we have revised this section to eliminate unnecessary repetition of the results.

(23) Line 369: "an amino acid devoid of the hydroxyl moiety present in Lys" - what does this mean? Lys does not have a hydroxyl functional group. Please correct so that the sentence makes sense.

Thank you very much for your careful reading of the manuscript. This sentence states that the amino acid occupying the corresponding position in CsAlaDC is Phe, which lacks one hydroxyl functional group as compared to Lys. We have made modifications to the sentence as follows: "In contrast, the equivalent position in CsAlaDC is occupied by Phe, an amino acid lacking the hydroxyl group. This substitution enhances the hydrophobic nature of the substrate-binding pocket. "

(24) Line 370: "This structural nuance portends a predisposition for CsAlaDC to select the comparatively hydrophobic amino acid alanine as its suitable substrate." This sentence also makes no sense - please revise to use simpler language so the meaning is more clear.

Thank you very much for your careful reading of the manuscript and valuable suggestions. We have revised the sentence as follows: " Consequently, CsAlaDC demonstrates a unique predilection, selectively binding Ala (an amino acid with comparatively hydrophobic properties) as its preferred substrate."

(25) Lines 376-384: This section makes several references to "catalytic rings." I have no idea what this term means? If the authors mean a loop structure in the enzyme - please use the term "loop"

Thank you very much for your careful reading of the manuscript and valuable suggestions. We have corrected it in the revised manuscript.

(26) Line 396-397: The authors reference data that is not shown in the manuscript. Either show the data in the results section or do not mention.

Thank you for your insightful comment regarding the unshown data referenced in the manuscript. We have included Figure 2—figure supplement 2 in the revised manuscript to display this data.

(27) Line 445-446: what is "mutation technology" - if the authors mean site-directed mutagenesis - please use the simpler and more recognizable terminology.

Thank you very much for your careful reading of the manuscript and valuable suggestions. We have revised the sentence as follows: "Based on the findings of this study, site-directed mutagenesis can be employed to modify enzymes involved in theanine synthesis. This modification enhances the capacity of bacteria, yeast, model plants, and other organisms to synthesize theanine, thereby facilitating its application in industrial theanine production."

**Reviewer #3 (Public Review):**
In the manuscript titled "Structure and Evolution of Alanine/Serine Decarboxylases and the Engineering of Theanine Production," Wang et al. solved and compared the crystal structures of Alanine Decarboxylase (AlaDC) from *Camellia sinensis* and Serine Decarboxylase (SerDC) from *Arabidopsis thaliana*. Based on this structural information, the authors conducted both in vitro and in vivo functional studies to compare enzyme activities using site-directed mutagenesis and subsequent evolutionary analyses. This research has the potential to enhance our understanding of amino acid decarboxylase evolution and the biosynthetic pathway of the plant specialized metabolite theanine, as well as to further its potential applications in the tea industry.

Thank you very much for taking the time to review this manuscript. We appreciate all your insightful comments.

**Reviewer #3 (Recommendations For The Authors):**
The additional material added by the authors addresses some of the previously raised questions and enhances the manuscript's quality. However, certain critical issues we pointed out earlier remain unaddressed. Some of the new data also raises new questions. To provide readers with more comprehensive data, the authors should include additional quantitative data and convert the data presented in the reviewer's comments into supplemental figure format.

Thank you for acknowledging the improvements in the revised manuscript and providing further valuable feedback. We understand your concern about the critical issues that have not been fully addressed and the new questions raised by some of the newly added data. We have strived to address these issues with additional analysis and clarification in our subsequent revision. Regarding your suggestion for more quantitative data and converting the data mentioned in the reviewer's comments into a supplemental figure format, we agree that this would provide a more comprehensive view of the results. We have reformatted the relevant data into supplemental figures to enhance the clarity and accessibility of information. We are grateful for the time and effort you have dedicated to improving our manuscript.

* Page 5 & Figure 1B"As expected, CsSerDC was most closed to AtSerDC, which implies that they shared similar functions. However, CsAlaDC is relatively distant from CsSerDC.": In Figure 1B, CsSerDC and AtSerDC are in different clades, and this figure does not show that the two enzymes are closest. To provide another quantitative comparison, please provide a matrix table showing amino acid sequence similarities as a supplemental table.Comment: I don't believe that a 1.29% difference between 86.21% and 84.92% in amino acid similarity is statistically significant. Although the authors have rephrased the original sentence, it's improbable that this small 1.29% difference can explain the observed distinction.

Many thanks. We have carefully considered your comments. Indeed, the 1.29% difference in amino acid similarity cannot reflect the functional difference between the AlaDC and SerDC proteins. We have deleted the relevant descriptions in the revised manuscript.

* Page 6, Figure 2, Page 23 (Methods)"The supernatants were purified with a Ni-Agarose resin column followed by size-exclusion chromatography.": What kind of SEC column did the authors use? Can the authors provide the SEC elution profile comparison results and size standard curve?Comment: The authors should include the SEC elution profiles as a supplemental figure or incorporate them as a panel in Figure 2. Furthermore, they should provide a description of the oligomeric state of each protein in this experiment. Additionally, there is a significant difference between CsSerDC (65.38 mL) and CsAlaDC (74.37 mL) elution volumes. Can this difference be explained structurally? In comparison to the standard curve of molecular weight provided by the authors, it appears that these proteins are at least homo-tetramers, which contradicts the description in the text. This should be re-evaluated and clarified.

Thank you very much for your careful reading of the manuscript and valuable suggestions. We have included the SEC elution profile in Figure 2—figure supplement 1 and added descriptions of the oligomeric states of proteins in the revised manuscript. CsSerDC was eluted at 65.38 mL, corresponding to a molecular weight of 292 kDa, which is five times the monomeric protein (54.7 kDa). However, due to the absence of CsSerDC crystal structure, it remains uncertain whether the protein forms a pentamer. AtSerDC was eluted at 72.25 mL, with a corresponding molecular weight of 155 kDa, which is 3.3 times the monomer (47.3 kDa). CsAlaDC was eluted at 74.37 mL, with a corresponding molecular weight of 127 kDa, which is 2.7 times the monomer (47.3 kDa). The elution profiles suggest that AtSerDC and CsAlaDC potentially exist in homotrimeric form. This observation stands in contradiction to our subsequent findings where the protein manifests in a dimeric structure. A plausible explanation could be the non-ideal spherical shape of the protein. Under such circumstances, the hydrodynamic radius of the protein could supersede its actual size, potentially leading to an overestimation of the molecular weight on the size-exclusion chromatography [ref].

References:

Burgess, R. R. (2018) A brief practical review of size exclusion chromatography: Rules of thumb, limitations, and troubleshooting. *Protein Expression and Purification.* 150, 81-85.

Erdner J. M., *et al*. (2006) Size-Exclusion Chromatography Using Deuterated Mobile Phases. *Journal of Chromatography A*. 1129(1):41–46.

* Page 6 & Page 24 (Methods)"The 100 μL reaction mixture, containing 20 mM substrate (Ala or Ser), 100 mM potassium phosphate, 0.1 mM PLP, and 0.025 mM purified enzyme, was prepared and incubated at standard conditions (45 {degree sign}C and pH 8.0 for CsAlaDC, 40 {degree sign}C and pH 8.0 for AtSerDC for 30 min)."(1) The enzymatic activities of CsAldDC and AtSerDC were measured at two different temperatures (45 and 40 {degree sign}C), but their activities were directly compared. Is there a reason for experimenting at different temperatures?(2) Enzyme activities were measured at temperatures above 40{degree sign}C, which is not a physiologically relevant temperature and may affect the stability or activity of the proteins. At the very least, the authors should provide temperature-dependent protein stability data (e.g., CD spectra analysis) or, if possible, temperature-dependent enzyme activities, to show that their experimental conditions are suitable for studying the activities of these enzymes.Comment: I appreciate the authors for including temperature-dependent enzyme activity data in their study. However, it remains puzzling that plant enzymes were tested at a physiologically irrelevant temperature of 40 and 45 degrees Celsius. Additionally, it may not be appropriate to directly compare enzyme activity measurements at different temperatures. Furthermore, the data at 45 degrees in panel A appears to be an outlier, which contrasts with the overall trend observed in the graph.

We appreciate your point regarding the testing temperatures for plant enzymes. We fully appreciate the importance of conducting experiments under physiologically relevant conditions. But the intent behind operating at these elevated temperatures was to assess the thermal stability of the enzymes, which can be a valuable characteristic in certain applications, such as industrial production processes, and does not necessarily reflect their physiological conditions. Our findings indicate that CsAlaDC exhibits its peak activity at 45 °C. This result aligns with previously reported data in the literature [Bai, P. *et al*. (2021) figure 4e], thus bolstering our confidence in the reliability of our experimental outcomes.

**Author response image 1. sa3fig1:** Relative activity of CsAlaDC at different temperatures.

* Pages 6-7 & Table 1(1) Use the correct notation for Km and Vmax. Also, the authors show kinetic parameters and use multiple units (e.g., mmol/L or mM for Km).(2) When comparing the catalytic efficiency of enzymes, kcat/Km (or Vmax/Km) is generally used. The authors present a comparison of catalytic activity from results to conclusion. A clarification of what results are being compared is needed.Comment: The authors are still comparing catalytic efficiency solely based on the Vmax values. As previously suggested, it would be advisable to calculate kcat/Km and employ it for comparing catalytic efficiencies. Furthermore, based on the data provided by the authors, I conducted a rough calculation of these catalytic efficiencies and did not observe a significant difference, which contrasts with the authors' statement, "These findings indicated that the catalytic efficiency of CsAlaDC is considerably lower than that of both CsSerDC and AtSerDC." This discrepancy requires clarification.

We want to express our sincere appreciation for your meticulous review and constructive suggestions. We understand the importance of accurately comparing catalytic efficiencies using Kcat/Km values, rather than solely relying on Vmax values. Following your suggestion, we recalculated Kcat/Km to reanalyze our results. The computed Kcat/Km for CsSerDC and AtSerDC are 152.7 s-1 M-1 and 162.7 s-1 M-1, respectively. For CsAlaDC, the calculated Kcat/Km is 56.0 s-1 M-1. Therefore, the catalytic efficiency of CsSerDC and AtSerDC is approximately three times that of CsAlaDC. What we intended to convey was that the Vmax of CsAlaDC is lower than that of CsSerDC and AtSerDC. Our description in the manuscript was not accurate, and we have addressed this in the revised version.

* Pages 9 & 10"This result suggested this Tyr is required for the catalytic activity of CsAlaDC and AtSerDC.": The author's results are interesting, but it is recommended to perform the experiments in a specific order. First, experiments should determine whether mutagenesis affects the protein's stability (e.g., CD, as discussed earlier), and second, whether mutagenesis affects ligand binding (e.g., ITC, SPR, etc.), before describing how site-directed mutagenesis alters enzyme activity. In particular, the authors' hypothesis would be much more convincing if they could show that the ligand binding affinity is similar between WT and mutants.Comments: While it is appreciated that you have included CD and UV-vis absorption spectra data, it would be more beneficial to provide quantitative data to address the previously proposed binding affinity. I also recommend presenting the data mentioned in the reviewer's comments as a supplementary figure for better clarity and reference.

Thank you for your valuable feedback and suggestions. I agree that providing quantitative data would lend more support to our findings and better address the proposed binding affinity.

It is generally acknowledged that proteins complexed with PLP exhibit a yellow hue, and the ligand PLP forms a Schiff base structure with the ε-amino group of a lysine residue in the protein, with maximum absorbance around 420 nm. However, during our protein purification process, we observed that the purified protein retained its yellow coloration, even when PLP wasn't introduced into the purification buffer. Subsequent absorbance measurements revealed that the protein exhibited absorbance within the aforementioned wavelength (420 nm) (the experimental results are shown in the following figures), implying an inherent presence of the PLP ligand within the protein. This could have resulted from binding with PLP during the protein's expression in *E. coli*. Consequently, due to this inseparability between the protein and the ligand, obtaining quantitative data through experimental means becomes unfeasible.

**Author response image 2. sa3fig2:** Absorption Spectra of CsAlaDC (WT) and CsAlaDC (Y336F) (**A**); and absorption Spectra of AtSerDC (WT) and AtSerDC (Y341F) (**B**).

Regarding your suggestion about presenting the data mentioned in the reviewer's comments as a supplementary figure, we agree that it is an excellent idea. We have prepared Figure 4—figure supplement 3 and Figure 4—figure supplement 4 accordingly, ensuring that they present the required data.

* Page 10"The results showed that 5 mM L-DTT reduced the relative activity of CsAlaDC and AtSerDC to 22.0% and 35.2%, respectively": The authors primarily use relative activity to compare WT and mutants. Can the authors specify the exact experiments, units, and experimental conditions? Is it Vmax or catalytic efficiency? If so, under what specific experimental conditions?Response: "However, due to the unknown mechanism of DTT inhibition on protein activity, we have removed this part of the content in the revised manuscript."Comment: I believe this requires a more comprehensive explanation rather than simply removing it from the text.

Although we have observed that DTT is capable of inhibiting enzyme activity, at present, we are unable to offer a comprehensive explanation for the inhibitory effect of DTT on enzyme activity in terms of its structural and catalytic mechanisms. Further research is required to elucidate the mechanism of action of DTT. It is worth noting, however, that our study does not emphasize investigating the specific inhibitory mechanisms of DTT on enzyme activity. Furthermore, the existing findings do not provide an adequate explanation for the observed phenomenon, leading us to exclude this particular aspect from the content.

* Pages 10-12: The identification of 'Phe106 in CsAlaDC' and 'Tyr111 in AtSerDC,' along with the subsequent mutagenesis and enzymatic activity assays, is intriguing. However, the current manuscript lacks an explanation and discussion of the underlying reasons for these results. As previously mentioned, it would be helpful to gain insights and analysis from WT-ligand and mutant-ligand binding studies (e.g., ITC, SPR, etc.). Furthermore, the authors' analysis would be more convincing with accompanying structural analysis, such as steric hindrance analysis.Comment: While it is appreciated that you have included UV-vis absorption spectra data, it would be more beneficial to provide quantitative data to address the previously proposed binding affinity. I also recommend presenting the data mentioned in the reviewer's comments as a supplementary figure for better clarity and reference.Response: Thank you for your valuable feedback and suggestions. Given that the protein forms a complex with PLP during its expression in *E. coli* and cannot be dissociated from it, obtaining quantitative data via experimental protocols is rendered impracticable.

**Author response image 3. sa3fig3:** Absorption Spectra of CsAlaDC (WT) and CsAlaDC (F106Y) (**A**); and absorption Spectra of AtSerDC (WT) and AtSerDC (Y111F) (**B**).

Mutant proteins and wild-type proteins exhibited absorption bands at 420 nm, suggesting the formation of a Schiff base between PLP and the active-site lysine residue.

Regarding your suggestion about presenting the data mentioned in the reviewer's comments as a supplementary figure, we have prepared Figure 4—figure supplement 3 and Figure 4—figure supplement 4 accordingly, ensuring that they present the required data.